# Monocarboxylate Transporter 4 Regulates Glioblastoma Motility and Monocyte Binding Ability

**DOI:** 10.3390/cancers12020380

**Published:** 2020-02-07

**Authors:** Sheng-Wei Lai, Hui-Jung Lin, Yu-Shu Liu, Liang-Yo Yang, Dah-Yuu Lu

**Affiliations:** 1Graduate Institute of Basic Medical Science, China Medical University, Taichung 40402, Taiwan; wayson081024@gmail.com; 2Department of Pharmacology, School of Medicine, China Medical University, Taichung 40402, Taiwan; linzoe9260@gmail.com (H.-J.L.); yushuliu220@gmail.com (Y.-S.L.); 3Department of Physiology, School of Medicine, China Medical University, Taichung 40402, Taiwan; 4Laboratory for Neural Repair and Research Center for Biotechnology, China Medical University Hospital, Taichung 40447, Taiwan; 5Department of Photonics and Communication Engineering, Asia University, Taichung 41354, Taiwan

**Keywords:** glioblastoma, hypoxic conditions, acidic microenvironment, MCT4, monocytes

## Abstract

Glioblastoma (GBM) is characterized by severe hypoxic and acidic stress in an abnormal microenvironment. Monocarboxylate transporter (MCT)4, a pH-regulating protein, plays an important role in pH homeostasis of the glycolytic metabolic pathways in cancer cells. The present study showed that GBM exposure to hypoxic conditions increased MCT4 expression. We further analyzed the glioma patient database and found that MCT4 was significantly overexpressed in patients with GBM, and the MCT4 levels positively correlated with the clinico-pathological grades of gliomas. We further found that MCT4 knockdown abolished the hypoxia-enhanced of GBM cell motility and monocyte adhesion. However, the overexpression of MCT4 promoted GBM cell migration and monocyte adhesion activity. Our results also revealed that MCT4-regulated GBM cell motility and monocyte adhesion are mediated by activation of the serine/threonine-specific protein kinase (AKT), focal adhesion kinase (FAK), and epidermal growth factor receptor (EGFR) signaling pathways. Moreover, hypoxia mediated the acetylated signal transducer and activator of transcription (STAT)3 expression and regulated the transcriptional activity of hypoxia inducible factor (HIF)-1α in GBM cell lines. In a GBM mouse model, MCT4 was significantly increased in the tumor necrotic tissues. These findings raise the possibility for the development of novel therapeutic strategies targeting MCT4.

## 1. Introduction

Glioblastoma (GBM) is the most common primary malignant brain tumor of all primary brain and central nervous system neoplasms [1,2]. Upon initial diagnosis of GBM, standard treatment consists of maximal surgical resection, radiotherapy, and adjuvant chemotherapy with temozolomide [3]. The highly heterogeneous tumor microenvironment plays a substantial role in treatment responses [4]. Histological studies have shown that GBM is one of the most hypoxic and angiogenic tumors [5,6]. GBM relies on robust intra-tumoral oxygenation and a pH-regulating system that leads to hypoxia and acidosis and the ability of tumor cells to adapt to the hypoxic microenvironment [7,8], which offers a distinct evolutionary advantage towards an aggressive phenotype [9,10].

Molecular mechanism responses to hypoxia are predominantly mediated by the transcriptional factor, hypoxia inducible factor (HIF)-1α [11]. HIF-1α is involved in the adaptive response under hypoxic conditions and in the regulation of many pivotal pathways in cancer [12,13]. Many recent studies have provided convincing evidence of strong correlations between elevated levels of HIF-1α and tumor metastasis, angiogenesis, and tumor resistance to therapy [14,15,16]. Emerging evidence suggests that HIF-1α inhibitors used in combination with standard chemotherapy improve the response of TMZ-resistant GBM cells [17]. Furthermore, silencing HIF-1α expression appears to inhibit the proliferation [18], invasion, and migration of GBM [19]. Importantly, overexpression of HIF-1α has been positively correlated with tumor progression and poor prognosis in patients with GBM [20,21].

Tumor cells produce increased amounts of H^+^ including lactic and carbonic acids, owing to the enhanced metabolic rates [22,23]. In addition, many studies point to tumor hypoxia increasing the expression of pH-regulating proteins that leads to acidosis of tumor microenvironments [24,25]. The resultant increased H^+^ production, coupled with poor extracellular clearance, creates a hypoxic and acidic niche that could rapidly become lethal to cells [26]. The bi-directional monocarboxylate transporters (MCTs) perform H^+^-linked transport of l-lactate across the plasma membrane and thus contribute to pHi regulation [27,28]. The MCT family consists of fourteen members [29,30]; MCT4 is the only member with H^+^-dependent transporters of monocarboxylic acids [31,32]. Increasing reports show that expression of MCT4 has also been linked to poor prognosis in several cancers [33,34]. Malignant tumors overexpress MCT4, characteristic of metastatic cancer in association with HIF-1α upregulation [35,36]. It has been shown that pharmacological or genetic ablation of MCT4 activity leads to reduced lung cancer cell proliferation [37]. Moreover, conditional MCT4 depletion efficiently inhibits tumor growth of glioma cancer stem cell (GSC)-derived xenografts [38]. Importantly, high expression of MCT4 has been positively correlated with tumor progression and poor prognosis in patients with GBM [39].

The present study showed that the activation of the epidermal growth factor receptor (EGFR) modulated HIF-1α and MCT4 expression under hypoxic conditions. In addition, hypoxia-induced acetylation of the signal transducer and activator of transcription (STAT3) contributed to HIF-1α and MCT4 expression, leading to GBM migration and monocyte adhesion. Furthermore, GBM-bearing mice also showed high expression of MCT4 in the tumor necrotic areas of GBM.

## 2. Results

### 2.1. MCT4 Expression Correlates with Pathological Grades of Human Glioma

It has been reported that hypoxia-induced HIF-1α is a key factor in modulating the expression of pH-regulating proteins [40]. As shown in Figure 1A, hypoxia induces HIF-1α and MCT4 protein expression in a time-dependent manner in human GBM. Similar effects were observed in the mouse GBM ALTS1C1 (Figure 1A). Hypoxia further induced MCT4 (SLC16A3), but not HIF-1α (HIF-1A) mRNA expression in a time-dependent manner (Figure 1B,C). pH measurement revealed significant acidification of the medium by GBM cells grown under hypoxic conditions (Figure 1D). We further analyzed the GSE4290 dataset showing that the mRNA levels of MCT4 were higher in the GBM group than in the astrocytoma groups (grade II and III). In addition, MCT4 levels were significantly higher in the GBM group than in the non-tumor group (Figure 1E).

### 2.2. MCT4 Is Involved in the Hypoxia-Enhanced GBM Migration and Monocyte Adhesion

It has been reported that MCT4 expression is correlated with malignancy in gliomas [39], and MCT4 elevation in GBM was also correlated with poor prognosis [41]. Based on a previous study and our earlier finding, we further investigated the role of MCT4 in the modulation of hypoxia in GBM. MCT4 mediated the binding of monocytes to GBM as determined by the monocyte-binding assay. As shown in Figure 2A–C, treatment of GBMs with the MCT4 inhibitor, CHC (a-cyano-4-hydroxycinnamic acid), decreased hypoxia-enhanced monocyte adhesion (green color; Figure 2A). In addition, a transwell assay was performed to further examine whether MCT4 facilitated GBM migration under hypoxic conditions. Treatment with CHC decreased the hypoxia-enhanced GBM migration activity in both of the human GBM cell lines, i.e., U87 and U251 (Figure 2D–F).

Similar effects were observed using the genetic method, wherein the transfection with shRNA against MCT4 attenuated the hypoxia-enhanced monocyte adhesion (Figure 3A–C) and GBM migration activity (Figure 3D–F). The most aggressive cancers rely on a robust glycolysis system leading to increased formation of intracellular lactate, which is exported to the extracellular environment by MCT4 [42]. 

To evaluate the direct effects of MCT4 in hypoxia-induced extracellular acidosis, we showed that knockdown of MCT4 by shRNA reduced the acidified tumor medium (Appendix A). However, transfection with the wild-type of MCT4 increased the GBM monocyte adhesion and migration activity in human GBM U87 and U251 cells (Figure 4A,B, Appendix A). These results indicated that hypoxia-induced MCT4 expression markedly enhanced GBM-associated monocyte adhesion and GBM motility.

### 2.3. HIF-1α Is Involved in the Hypoxia-Induced MCT4 Expression in GBM

Next, we further examined whether HIF-1α modulated MCT4 expression under hypoxic conditions. As shown in Figure 5A, GBM treated with the HIF-1α inhibitor showed decreased hypoxia-induced MCT4 expression. Additionally, GBM was first exposed to hypoxia-induced MCT4 transcription activity and then treated with the HIF-1α inhibitor, which reversed the hypoxia-enhanced MCT4 expression (Figure 5B). Moreover, GBM was treated with an HIF-hydroxylase inhibitor DMOG (dimethyloxalylglycine) which can cause HIF-1α protein stabilization. As shown in Figure 5C, DMOG dramatically increased HIF-1α and MCT4 protein expressions under normoxia. We further analyzed GSE4290, of which the dataset showed that the levels of HIF-1α were higher in the GBM group than in the astrocytoma groups (grade II and III). In addition, HIF-1α levels were significantly higher in the GBM group than in the non-tumor group (Figure 5D). Notably, Pearson’s correlation analysis also showed a positive correlation between HIF-1α and MCT4 expression levels in the gene expression dataset of patients with glioma (Figure 5E). These data suggest that hypoxia stimulates HIF-1α protein stabilization and subsequently increases MCT4 expression.

### 2.4. Acetylated STAT3 Is Involved in Hypoxia-Induced HIF-1α and MCT4 Expression in GBM

A previous study reported that tumor cell responses to hypoxia are predominantly mediated by HIF-1α, which transactivates more genes essential for adaptation and tumor survival [43,44]. As shown in Figure 6A, human GBM cells exposed to hypoxia resulted in the translocation of HIF-1α from the cytoplasm to the nucleus. Moreover, GBM cells exposed to hypoxia showed increased HIF-1α binding to the hypoxia response element (HRE) binding site on the *MCT4* promoter (Figure 6B). On the other hand, the histone deacetylase (HDAC) inhibitor and histone acetyl transferases (HATs) are believed to promote transcription by enhancing acetylation of histones, transcription factors, and coactivators [45,46]. As shown in Figure 6C, treatment with the HDAC inhibitor (SAHA) increased hypoxia-induced HIF-1α and MCT4 protein expressions in GBM cells. In addition, GBM cells incubated under hypoxic conditions showed increased STAT3 acetylation in a time-dependent manner (Figure 6D). Furthermore, inhibition of STAT3 by a STAT3 pharmacological inhibitor, stattic, effectively antagonized the hypoxia-induced HIF-1α and MCT4 expressions in the human GBM U87 and U251 cell lines (Figure 6E). The above results indicated that post-translational modification of STAT3 was involved in the hypoxia-induced HIF-1α and MCT4 expressions in GBM.

### 2.5. Activation of FAK and AKT is Involved in Hypoxia-Induced HIF-1α and MCT4 Expression in GBM

Recent evidence suggests that focal adhesion kinase (FAK) and serine/threonine-specific protein kinase (AKT) could be dual kinase targets that prevent cancer cell adhesion and metastasis [47]. Our previous study indicated that bradykinin-induced IL-8 expression and GBM migration are mediated by the FAK/STAT3 signaling pathways [48]. Moreover, hypoxia increased cancer cell migration via HIF-mediated FAK phosphorylation [49]. As shown in Figure 7A, GBM cells exposed to hypoxic conditions increased FAK and AKT phosphorylation in a time-dependent manner. Next, we further investigated whether FAK and AKT were involved in HIF-1α-mediated MCT4 expression. Treatment with a FAK pharmacological inhibitor, PF573228, reduced the hypoxia-induced HIF-1α and MCT4 expression in GBM (Figure 7B). Moreover, treatment with the PI3 kinase/AKT pharmacological inhibitor LY294002 also attenuated the hypoxia-induced HIF-1α and MCT4 expression in GBM cells (Figure 7C). On the other hand, treatment with FAK or PI3 kinase/AKT inhibitors effectively reduced hypoxia-enhanced MCT4 transcription activity (Figure 7D). These results indicate that FAK and AKT signaling is an important effector of hypoxia-mediated stabilization of the expression of the HIF-1α protein and MCT4 in GBM.

### 2.6. Activation of EGFR Is Involved in Hypoxia-Induced HIF-1α and MCT4 Expression in GBM

It has been reported that amplification of EGFR and its active mutants occurs frequently in GBM [50] and is associated with a poor patient outcome and survival rate [51]. Our previous study also showed that STAT3 signaling was involved in the EGFR-associated adhesion molecule expression and monocyte adhesion in GBM [50]. We now investigated whether EGFR is involved in HIF-1α-mediated MCT4 expression. Treatment with EGFR tyrosine kinase inhibitors, erlotinib, gefitinib or AG1478, reduced the hypoxia-induced MCT4 expression in GBM (Figure 8A,B). Similarly, treatment with EGFR inhibitors significantly reduced hypoxia-enhanced MCT4 transcription activity (Figure 8C,D). These results indicated that EGFR signaling was an important effector of hypoxia-mediated stabilization of the HIF-1α protein and MCT4 expression in GBM.

### 2.7. Expression of MCT4 Is Elevated in the GBM Mouse Model

Based on the strong correlation between MCT4 expression and poor outcomes in GBM, we further determined whether MCT4 was a modulator of aggressive cancer in an in vivo study. Mice were intracranially injected with mouse glioma ALTS1C1. Tumors were allowed to grow, and all animals were sacrificed 14 days after tumor implantation. Figure 9A shows the illustration of different brain regions we used in this study. Analysis of the tumor in the ALTS1C1-bearing mice showed that protein (Figure 9B,C) or gene (Figure 9D) expression levels of MCT4 were higher in tumor necrotic tissues compared with the adjacent tissues. Furthermore, MCT4 expression was further assessed by immunohistochemical staining and was found to be increased in the tumor necrotic areas of mice with GBM (Figure 9E). These results indicated that MCT4 levels were significantly correlated with the progression of GBM.

## 3. Discussion

Recent studies have focused on targeting the tumor microenvironment for anti-cancer therapies and for identifying a valuable biomarker for prognostic purposes. The present study demonstrated the differential expression of the lactate transporter MCT4 in GBM under hypoxia. DNA microarray (GEO dataset) analysis also indicated that the higher expression of MCT4 in patients with glioma correlated with a poor prognosis. Our finding also showed a positive correlation of MCT4 expression with GBM motility and monocyte adherent abilities of GBM. In addition, overexpression of MCT4 increased GBM motility and monocyte adhesion under normoxia conditions. Moreover, administration of the HIF-1α inhibitor effectively reduced MCT4 expression. Furthermore, Pearson’s correlation analysis showed a positive correlation between MCT4 and HIF-1α expression levels in glioma patients. Using chromatin immunoprecipitation (ChIP) assays, we demonstrated that HIF-1α binding to the hypoxia response element (HRE) binding site on the MCT4 promoter under hypoxia was regulated by acetylation of STAT3. Moreover, inhibition of STAT3 effectively reduced hypoxia-induced HIF-1α and MCT4 expression. Finally, EGFR inhibitors erlotinib, gefitinib, and AG1478 effectively reduced the enhancement of HIF-1α and MCT4 expression as well as MCT4 transcriptional activity under hypoxic conditions. Our study suggests that hypoxia-induced MCT4 expression enhances GBM progression by increasing GBM motility and monocyte recruitment (Figure 9F). 

Lactic acidosis and hypoxia are two prominent microenvironmental stresses in solid tumors [52]. The present study suggests that MCT4 expression positively correlates with glioma grade and poor prognosis of patients with glioma (Figure 1D). Recently, increasing evidence has shown that lactate presence in the extracellular space, which is mediated by MCTs, has strong immunosuppressive effects [25]. In addition, MCT4 modulates macrophages into an M-2-like state that suppresses the inflammatory response and inhibits monocyte differentiation [53,54,55]. It has been demonstrated that classical MCT inhibitors and knockdown of MCT4 reduce the migration and invasion capacity of breast, lung, and glioma cells [53,54,56,57]. The present study also revealed that the overexpression of MCT4 was closely associated with monocyte adhesion (Figure 4A) and cell motility (Figure 4B). Our results showed that pharmacological (Figure 2A,B) or genetic (Figure 3A,B) inhibition of MCT4 significantly reduced monocyte adhesion ability. GBM exhibited increased expression levels of the *MCT* genes *SLC16A1* and *SLC16A3* when compared to normal brain parenchyma, as well as oligodendrogliomas and astrocytomas [39]. It has been reported that MCT4 is expressed in few cases of nonneoplastic brain tissues [58]. Brain samples from the GBM mouse model showed higher levels of MCT4 in tumor necrotic tissues compared to the adjacent tissues (Figure 9C,D). These findings suggest that MCT4 probably represents a critical therapeutic target for GBM and could serve as a prognostic indicator.

Post-translational modification is one of the mechanisms by which protein function is regulated by chronic hypoxia [59]. HIF-1α is one of the most important regulators of dynamic balance between histone acetyl transferases (HATs) and histone deacetylases (HDACs) [60]. Hypoxia-mediated histone acetylation and expression of N-myc transcription factor dominate aggressiveness of neuroblastoma cells [61]. Our results showed that HDAC inhibitors augmented hypoxia-induced HIF-1α and MCT4 protein expression in the GBM cells (Figure 6C). In our previous study, STAT3 levels correlated significantly with the clinico-pathological grade of glioma. Moreover, STAT3 phosphorylation and translocation to the nucleus promote GBM migration [48]. Previous reports have indicated that STAT3 increases HIF-1α protein stability and interacts with HIF-1α target gene promoters [62]. Other reports have also shown that activation of STAT3 is involved in the modulation of expression of pH-regulating proteins, CA9 and MCT4 [63]. The present study revealed that acetyl-STAT3 was activated in response to hypoxia, and inhibition of STAT3 decreased hypoxia-induced MCT4 and HIF-1α expression (Figure 6D,E). High levels of phosphorylated AKT have been reported to correlate with the poor prognosis of patients with GBM [64]. HIF-1α has been shown to lie downstream of EGFR and AKT signaling and to activate VEGF expression in tumor cells [65]. In addition, our previous study also supports the finding that hypoxia-induced iNOS expression in microglia is HIF-1α-dependent and involves the activation of the PI3-kinase/AKT/mTOR signaling pathway [66]. Another study revealed that AKT inhibitors decrease HIF-1α post-transcriptional protein levels, but not the transcriptional mRNA levels in the brain tissue [67]. However, FAK is typically considered as being upstream of AKT, and extracellular pressure stimulates cancer cell adhesion via AKT and FAK activation [47]. Over-activation of STAT3 is conducive to tumor invasiveness by up-regulation of FAK in gastric cancer cells [68]. Our present study found that FAK/STAT3 signaling is involved in GBM migration and IL-8 production [48]. In addition, numerous studies have suggested that EGFR is overexpressed in most primary GBM and is characteristic of more aggressive glioblastoma phenotypes [69,70]. Amplification of EGFR is associated with GBM proliferation and invasion [71]. Moreover, inhibition of GBM cells by EGFR is associated with anti-angiogenic and pro-apoptotic effects on the tumor [72]. It has been reported that coactivation of EGF and EGFR drives tumor metastasis via PI3K/AKT-dependent pathways [73]. Targeting FAK/STAT3 sensitizes gliomas to the anti-EGFR agent gefitinib and alkylating agent temozolomide in human gliomas [74]. Of note, small molecules against MCT4, as combination therapy with temozolomide, improve chemosensitivity and survival of GBM cells [58]. Our previous study has shown that STAT3 signaling is involved in the EGFR-associated adhesion molecule expression and monocyte adhesion in GBM [50]. Our results suggest that MCT4-regulated GBM motility and monocyte adhesion are mediated by activation of AKT, FAK, and EGFR signaling pathways. However, further research should explore the specific mechanism by which AKT, FAK, and EGFR/STAT3 affect the expression of MCT4 and should determine how pH-regulating proteins affect GBM and form a suppressive immune microenvironment.

## 4. Materials and Methods 

### 4.1. Materials

DMOG, AG1478 (EGFR inhibitor), SAHA (HDAC inhibitor), Trichostatin A (HDAC inhibitor), PF573228 (FAK inhibitor), and LY-294002 (PI3 kinase/AKT inhibitor) were obtained from Sigma–Aldrich (St. Louis, MO, USA). The HIF-1 inhibitor was obtained from Calbiochem (San Diego, CA, USA). Stattic (STAT3 inhibitor), erlotinib (EGFR inhibitor), and gefitinib (EGFR inhibitor) were obtained from Selleckchem (Houston, TX, USA). Primary antibodies against HIF-1α (610958) were obtained from BD Biosciences (San Jose, CA, USA). Primary antibodies against GAPDH (SI-G8795) and α-tubulin (SI-T5168) were obtained from Sigma–Aldrich. Primary antibodies against β-actin (sc-47778), PCNA (sc-56), AKT1 (sc-5298), p-AKT1/2/3 (sc-7985-R), FAK (sc-271195), and MCT4 (sc-376140) were obtained from Santa Cruz Biotechnology (Santa Cruz, CA, USA). Primary antibodies against Ac-STAT3 (2523), p-FAK (3283), and EGFR (4267) were obtained from Cell Signaling Technology (Danvers, MA, USA). On-Target smart pool MCT4 small interfering (si)RNA and control non-targeting siRNA were obtained from Dharmacon (Lafayette, CO, USA).

### 4.2. Animals

Male (8-week-old C57BL/6) mice were obtained from the National Laboratory Animal Center (Taipei, Taiwan) and were housed under standard laboratory conditions (21 ± 2 °C, 12 h L/D cycle, with food and water available ad libitum). All animal procedures were performed in accordance with the Animal Care and Use Guidelines of China Medical University (Taichung, Taiwan).

### 4.3. Cell Culture

Human glioma U251 cells were purchased from the Japanese Collection of Research Bioresources Cell Bank (JCRB NO. IFO50288, Japan). Human glioma U87 cells, mouse glioma ALTS1C1 cells, and human monocyte THP-1 were purchased from the Bioresource Collection and Research Center (BCRC No. 60360, 60582 and 60430; Taiwan). U251 and U87 cells were maintained with minimum essential medium (MEM), ALTS1C1 cells were maintained with Dulbecco’s modified eagle medium (DMEM), and THP-1 cells were maintained with RPMI-1640 medium. All the culture cells were grown in medium containing 10% fetal bovine serum (FBS), 100 mg/mL streptomycin, and 100 U/mL penicillin (PS). All the cells were incubated at 37 °C in a humidified atmosphere containing 5% CO_2_ and 95% air. For hypoxic exposure, cells were maintained at 37 °C in a sealed acrylic chamber flushed with a gas mixture containing 1% O_2_, 5% CO_2_, and 94% N_2_.

### 4.4. Cytosolic and Nuclear Extracts

Nuclear extracts were prepared as previously described [75]. Briefly, cells were rinsed with cold PBS and resuspended in a hypotonic buffer (10 mM HEPES, pH 7.6, 10 mM KCl, 1 mM dithiothreitol, 0.1 mM EDTA, and protease inhibitor cocktail) for 10 min on ice. The cytosolic proteins were separated by centrifugation at 10,000× *g* for 2 min. The supernatants containing the cytosolic proteins were collected, and the pellets containing the nuclear fraction were resuspended in buffer (20 mM HEPES pH 7.6, 1 mM EDTA, 1 mM dithiothreitol, 0.4 M NaCl, 25% glycerol, and protease inhibitor cocktail) for 30 min on ice. The suspensions were centrifuged again at 13,000× *g* for 20 min, and the supernatants containing the nuclear proteins were collected and stored at −80 °C.

### 4.5. Monocyte-Binding Assay

The procedure of the monocyte-binding assay was described previously [76,77]. Briefly, human monocyte THP-1 cells were incubated with BCECF/AM (2′,7′-bis-(2-carboxyethyl)-5-(and-6)-carboxyfluorescein) and 0.1 μg fluorescent dye of in RPMI-1640 medium in an incubator for 1 h. GBM cells were treated with CHC or were transfected with short hairpin or wild-type MCT4 for the different time periods. Then, the medium was removed from the wells, and a monolayer of GBM cells was added with 2.0 × 105 BCECF/AM-labeled-THP-1 cells to each well. Non-adherent monocytes were removed and gently washed twice with culture medium and incubated for 45 min. The adherent monocytes were then photographed and counted using a fluorescence microscope.

### 4.6. Western Blotting

Western blotting of whole cell extracts was performed according to a previous study [78]. Briefly, cells were extracted with RIPA lysis buffer; the cells were collected by a scraper and were kept on ice. The protein samples were spun at 13,000× rpm for 20 min, after which the supernatant was collected and stored at −20 °C. Protein samples were separated by sodium dodecyl sulfate-polyacrylamide gel electrophoresis and were then transferred onto PVDF membranes (Millipore, Bedford, MA, USA). Membranes were blocked by non-fat dry milk (5%) in TBST for 1 h. The membranes were incubated with primary antibodies at 4 °C overnight or with RT for 1 h. Following washes with TBST buffer, the membranes were incubated with anti-mouse or anti-rabbit HRP-conjugates secondary antibodies. Protein bands were visualized by ECL and Kodak X-OMAT LS film (Eastman Kodak, Rochester, NY, USA). The data were quantified using ImageJ software (National Institutes of Health, Bethesda, Maryland, USA, https://imagej.nih.gov/ij/, 1997–2018).

### 4.7. Reverse Transcription and Real-Time PCR

Total RNA was isolated from GBM cells by TRIzol (TRI Reagent), and the concentration of RNA was measured with a BioDrop spectrophotometer. The target gene expression was detected by quantitative real-time PCR (q-PCR). The messenger RNA converted into cDNA by reverse transcription (RT) using an Invitrogen Reverse Transcription Kit and was amplified using the oligonucleotide primers as follows: hHIF-1α: 5′-TTACA GCAGC CAGAC GATCA-3′ and 5′-CCCTG CAGTA GGTTT CTGCT- 3′; hMCT4: 5′-GAGTT TGGGA TCGGC TACAG-3′ and 5′-CGGTT CACGC ACACA CTG-3′, and internal control h36B4: 5′-AGATG CAGCA GATCC GCAT-3′ and 5′-GTTCT TGCCC ATCAG CACC-3′. The PCR reaction using SYBR Green Master Mix (Applied Biosystems, Foster City, CA, USA) was performed on a StepOne Plus Real-Time PCR System. The threshold was set within the linear phase of target gene amplification to calculate the cycle number at which the transcript was detected (denoted as CT).

### 4.8. Luciferase Reporter Assay

GBM cells were transfected with MCT4 firefly luciferase reporter plasmids and SV40-renilla luciferase control plasmids using Lipofectamine 3000 (Invitrogen) according to the manufacturer’s recommendations. After 24 h transfection, the cells were pretreated with inhibitors for 40 min and were then exposed to hypoxia for 24 h. Cell extracts were then prepared, and firefly and renilla luciferase activities were measured.

### 4.9. Cell Transfection

GBM cells were transiently transfected with wild-type MCT4 and MCT4 luciferase reporter plasmids using Lipofectamine (LF)3000 (Invitrogen) for 24 h. Plasmid DNA and LF3000 were premixed in serum-free medium for 5 min and were then applied to the cells. The LF3000-containing medium was replaced with fresh serum-free medium after 24 h. GBM cells were transiently transfected with shRNA against MCT4 or control shRNA (Dharmacon, Lafayette, CO, USA) using DharmaFECT transfection reagents (Dharmacon). The shRNA or the negative control were premixed with the DharmaFECT transfection reagent in a serum-free medium for 10 min and were then used to transfect the cells. After 24 h incubation, the medium containing Lipofectamine 3000 was replaced with fresh serum-free medium.

### 4.10. Cell Migration Assay

In vitro migration assay was performed using a Costar transwell insert (8-μm pore size; Costar, NY, USA) in 24-well plates as previously described [48]. Prior to the migration assay, the GBM cells were treated with CHC or were transfected with short hairpin or wild-type MCT4. Approximately 2 × 10^4^ GBM cells in 200 μL of serum-free medium were placed in the upper chamber, and 400 μL of the same medium was placed in the lower chamber. The plates were incubated for 24 h at 37 °C under hypoxic conditions (1% O_2_). The cells that migrated through the filters were fixed with 3.7% formaldehyde for 5 min and then stained with 0.05% crystal violet for 30 min. Cells on the upper side of the filters were wiped with cotton-tipped swabs, and the cells on the underside of the filter were photographed using a digital camera mounted on a microscope.

### 4.11. Chromatin Immunoprecipitation (ChIP) Assay

The ChIP assay was performed using an EZ-Magna ChIPTM A/G Chromatin Immunoprecipitation Kit (Millipore, Billerica, MA, USA) as previously described [50] using isolated nuclei from the formaldehyde cross-linked GBM cells. Immunoprecipitation was performed using the primary antibodies against HIF-1α and magnetic beads. Normal mouse IgG was used as a negative control, and 1 μg of antibody was used for each reaction. The diluted chromatin was then incubated on a rotator at 4 °C for overnight and then extracted and purified. Purified DNA was subjected to PCR amplification using the oligonucleotide primers, 5′-AACGC TCTGG TTGCA AATAA AA-3′ and 5′-ACGCA CTTGT AATTA CTCAA ACA-3′, which were used to amplify the MCT4 promoter region. The PCR products were resolved by 2% agarose gel electrophoresis and were visualized under UV light.

### 4.12. GEO Gene Expression Database

The DNA microarray data were obtained from the glioma patient dataset, accession number GSE4290. The expression levels of target genes were analyzing using GraphPad Prism 6 software from the publicly available Gene Expression Omnibus (GEO) databases. The glioma patients were collected from the Henry Ford Hospital (HFH) that contains 180 glioma patients with histologically confirmed different grades of glioma: grade IV astrocytomas (GBM *n* = 81), grade III (astrocytomas *n* = 19, oligodendrogliomas *n* = 12), grade II (astrocytomas *n* = 7, oligodendrogliomas *n* = 38), and non-tumors *n* = 23. The gene expression of HIF-1α and MCT4 was obtained from the GSE4290 dataset and correlated with the human glioma pathological grade.

### 4.13. Intracranial Tumor Implantation

ALTS1C1 was freshly prepared and adjusted to 1 × 10^5^ cells/mL before implantation. An intracranial injection was performed according to protocols approved by the China Medical University Animal Care and Use Committee. Briefly, C57BL/6 mice were anesthetized and placed in a stereotactic frame, and the skulls were exposed by incision. In each skull, a hole was made, 0.8 mm anterior, 2.5 mm to the right of the bregma, and cells (2 μL) were injected using a 10-μL Hamilton syringe with a 26S-gage needle mounted in a stereotactic holder. The syringe was lowered to a depth of 3 mm, and the cells were injected at a rate of 0.4 μL/min. After intracranial implantation, a 5-min waiting period was observed before slowly withdrawing the syringe to prevent any reflux. The skull was then cleaned, and the incision was sutured. Tumors were allowed to grow, and all animals were sacrificed 24 days after tumor implantation.

### 4.14. Immunohistochemistry

The protocol for immunohistochemistry was performed as described in our previous report [79]. Briefly, brain tumor tissues on glass slides were rehydrated and quenched for endogenous peroxidases with hydrogen peroxide. After being deparaffinized, the sections were blocked by incubation in PBS containing 3% bovine serum albumin. The anti-MCT4 primary antibody was applied to the slides at a dilution of 1:50, followed by overnight incubation at 4 °C. After several washes with PBS, the samples were incubated with the biotin-conjugated secondary antibody at a dilution of 1:50. Bound antibodies were detected with the ABC reaction kit (Vector Laboratories; Burlington, CA, USA), developed with diaminobenzene (Sigma–Aldrich), and counterstained with hematoxylin. The results were presented the mean ± S.E.M. and all the data were collected using at least three biologically independent replicates.

### 4.15. Statistics

The results are presented as the mean ± S.E.M., and all the data were collected based on at least three biologically independent replicates. The values were determined using ImageJ software and GraphPad Prism 8 software (version 8, GraphPad software Inc., San Diego, CA, USA). The data given are based on statistical analysis between two samples that was performed using a Student’s *t*-test. For multiple comparisons, one-way ANOVA or two-way ANOVA was performed, followed by the Bonferroni test. In all cases, a *p*-value < 0.05 was considered to be statistically significance. The *p*-values are indicated in the figure legends. No pre-test was used to choose the sample size.

## 5. Conclusions

In this study, we elucidated the mechanism by which EGFR and acetylated STAT3 stabilize HIF-1α and further regulate the expression of MCT4 and thus affect GBM motility and monocyte adhesion under hypoxia. In vivo studies suggested that MCT4 was up-regulated in the GBM mouse model. Of note, the MCT4 level was significantly increased in the tumor necrotic tissues of GBM. These findings raise the possibility that strategies targeting MCT4 might provide an opportunity for the development of novel therapeutic targets.

## Figures and Tables

**Figure 1 cancers-12-00380-f001:**
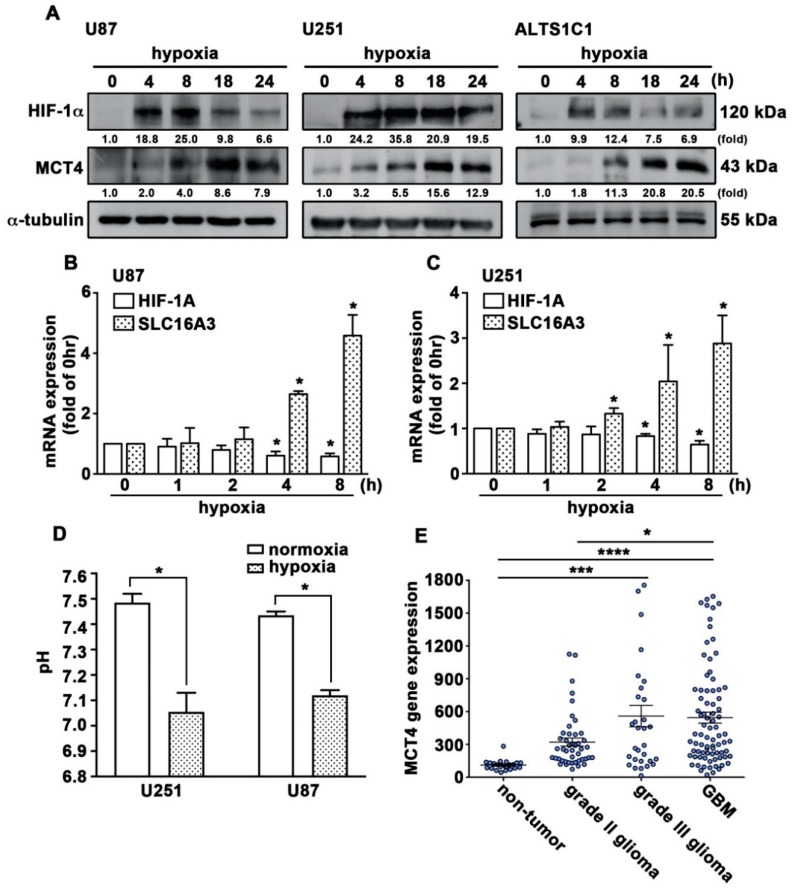
Hypoxia induces HIF-1α and MCT4 expression in GBM. (**A**) Human GBM cell lines U87 and U251 and the mouse GBM cell line ALTS1C1 were exposed to hypoxic conditions (1% O_2_) for indicated time periods (4, 8, 18, or 24 h). HIF-1α and MCT4 protein expressions were determined using western blotting. U87 (**B**) and U251 (**C**) were exposed to hypoxic conditions (1% O_2_) for indicated time periods (1, 2, 4, or 8 h), and *HIF-1α*, and *MCT4* mRNA expression was determined by qPCR. One-way ANOVA with a post-hoc Bonferroni test was used to examine the significance of the mean. * *p* < 0.05 compared with the control group. Quantitative data are presented as the mean ± S.E.M. (*n* = 3). (**D**) pH measurements of culture medium 24 h after growing GBM cells under hypoxic conditions (1% O_2_). * *p* < 0.05 compared with the normoxia group (Student’s *t*-test). Quantitative data are presented as the mean ± S.E.M. (*n* = 3). (**E**) mRNA levels of *MCT4* in patients’ specimens from the human glioma microarray dataset GSE4290. One-way ANOVA with a post-hoc Bonferroni test was used to examine the significance of the mean. * *p* < 0.05 GBM compared with grade II glioma. *** *p* < 0.001 grade III glioma compared with non-tumor. **** *p* < 0.0001 GBM compared with non-tumor.

**Figure 2 cancers-12-00380-f002:**
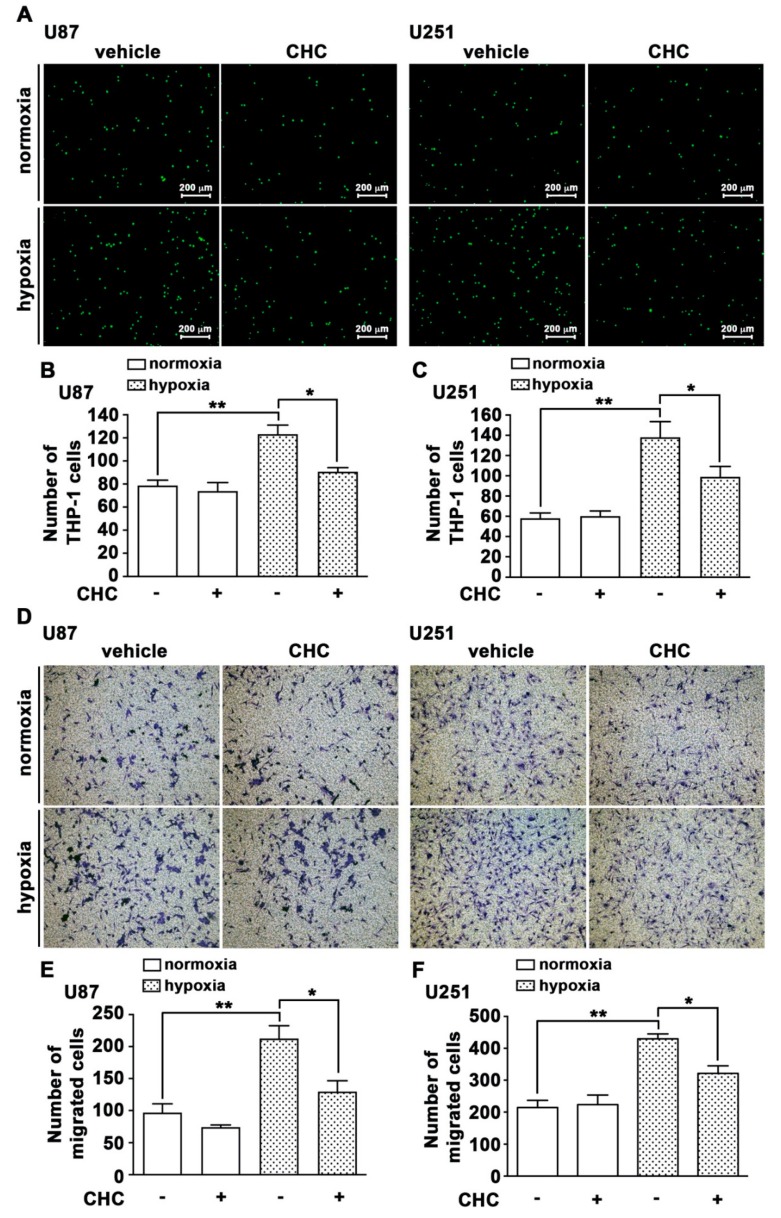
MCT4 is involved in hypoxia-enhanced monocyte adhesion and GBM migration. (**A**) U87 and U251 were treated with an MCT4 inhibitor (CHC; 1 or 2.5 mM) for 30 min and were exposed to hypoxic conditions (1% O_2_) for 24 h. BCECF-AM-labeled-THP-1 was added to U87 and U251 for 40 min, and then the adherence of THP-1 was captured by fluorescence microscopy. Quantification of THP-1 monocyte adhesion abilities on GBM U87 (**B**) or U251 (**C**). (**D**) GBM U87 and U251 were treated with an MCT4 inhibitor (CHC; 1 or 2.5 mM) for 30 min and were exposed to hypoxic conditions (1% O_2_), and the migration activities were measured after 24 h by a transwell assay and were visualized using a digital camera. Quantification of GBM U87 (**E**) and U251 (**F**) migration by number of cells that migrated to the underside of the membrane. Two-way ANOVA with a post-hoc Bonferroni test was used to examine the significance of the mean. * *p* < 0.05 compared with the hypoxia-only group. ** *p* < 0.01 compared with normoxia control group.

**Figure 3 cancers-12-00380-f003:**
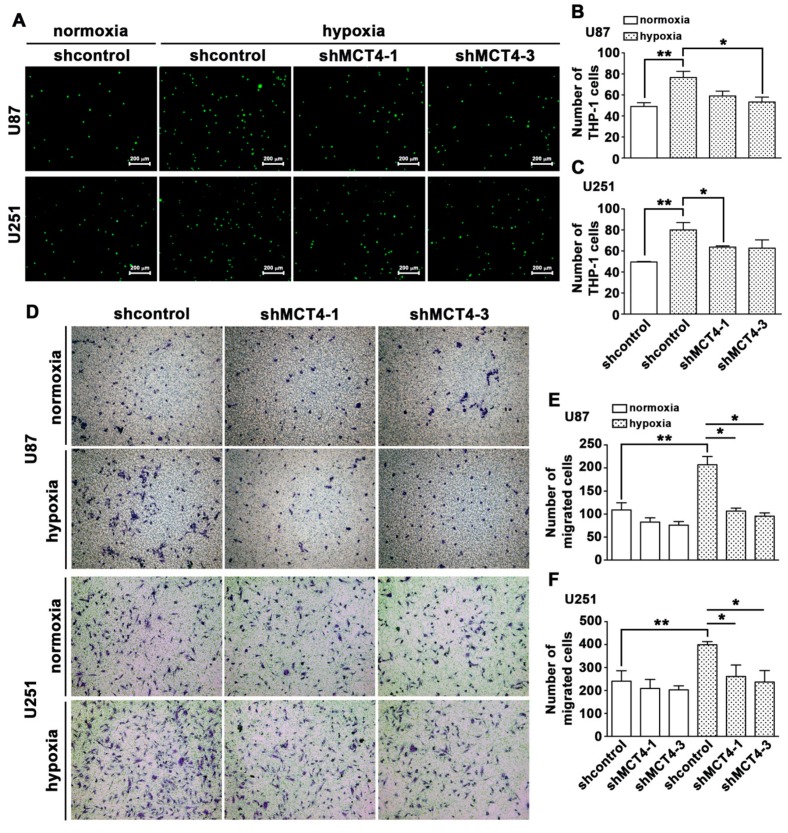
Downregulation of MCT4 attenuates hypoxia-enhanced monocyte adhesion and GBM migration. (**A**) U87 and U251 cells were transfected with the control or *MCT4* shRNA for 24 h and were exposed to hypoxic conditions (1% O_2_) for 24 h. BCECF-AM-labeled-THP-1 was added to U87 and U251 cells for 40 min, and then the adherence of THP-1 was measured by fluorescence microscopy. Quantification of monocyte adhesion abilities on GBM U87 (**B**) or U251 (**C**) cells. (**D**) GBM U87 and U251 cells were transfected with the control or *MCT4* shRNA for 24 h and were exposed to hypoxic conditions (1% O_2_) for 24 h; migration activities were measured by a transwell assay and were visualized using a digital camera. Quantification of U87 (**E**) and U251 (**F**) cell migration by counting the number of cells that migrated to the underside of the filter. Two-way ANOVA with a post-hoc Bonferroni multiple was used to examine the significance of the mean. * *p* < 0.05 compared with the hypoxia control (sh) group. ** *p* < 0.01 compared with the normoxia control (shcontrol) group.

**Figure 4 cancers-12-00380-f004:**
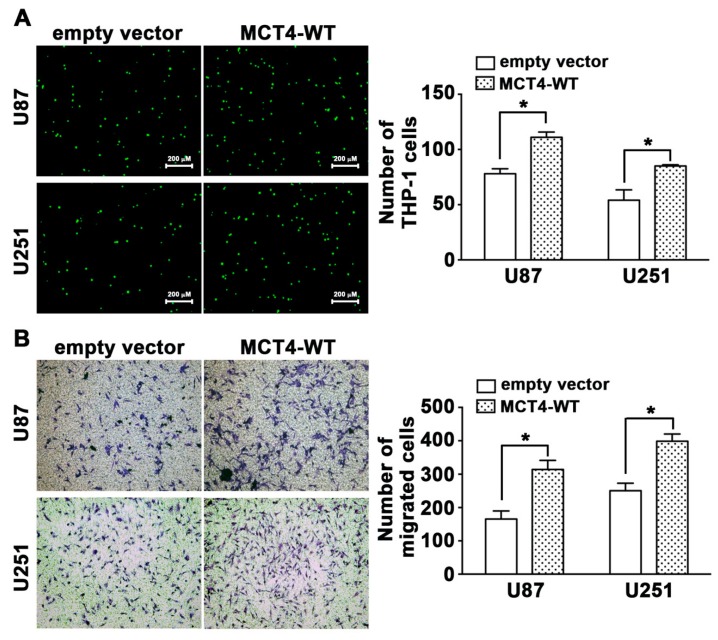
Overexpression of MCT4 enhances monocyte adhesion and GBM migration. U87 and U251 cells were transfected with either an empty vector or wild-type *MCT4* for 24 h and were exposed to hypoxic conditions (1% O_2_) for 24 h. (**A**) quantification of monocyte adhesion ability (right panel) was quantified using fluorescence microscopy. (**B**) GBM migration activity (right panel) was quantified using a digital camera. * *p* < 0.05 compared with the empty vector group (Student’s *t*-test). Quantitative data are presented as the mean ± S.E.M. (*n* = 3).

**Figure 5 cancers-12-00380-f005:**
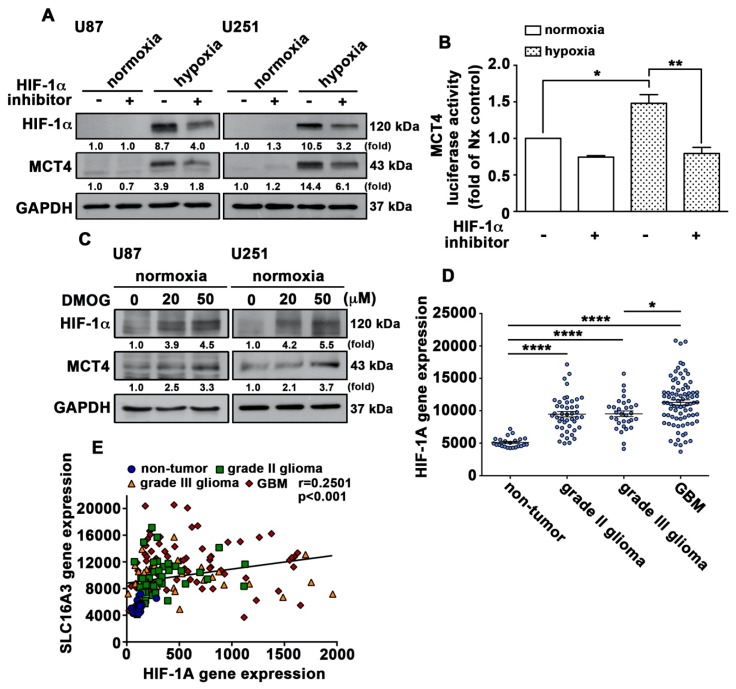
HIF-1α is involved in hypoxia-induced MCT4 expression. (**A**) U87 and U251 cells were treated with the HIF-1α inhibitor (10 μM) for 30 min and were exposed to hypoxic conditions (1% O_2_) for another 24 h. (**B**) U251 cells were transfected with MCT4 luciferase reporter plasmids for 24 h and were treated with the HIF-1α inhibitor (10 μM) for 30 min and were then exposed to hypoxic conditions (1% O_2_). Firefly luciferase activity was measured after 24 h, and the results were normalized to renilla luciferase activity. Two-way ANOVA with a post-hoc Bonferroni test was used to examine the significance of the mean. * *p* < 0.05 compared with the normoxia control group. ** *p* < 0.01 compared with the hypoxia-only group. Quantitative data are presented as the mean ± S.E.M. (*n* = 3). (**C**) U87 and ALTS1C1 cells were treated with DMOG (20, or 50 μM) for 24 h. HIF-1α and MCT4 protein expressions were determined using western blotting. (**D**) mRNA levels of *HIF-1α* in patient specimens from the human glioma microarray dataset GSE4290. One-way ANOVA with a post-hoc Bonferroni test was used to examine the significance of the mean. * *p* < 0.05 GBM compared with grade III glioma. **** *p* < 0.0001 compared with non-tumor. (**E**) Pearson’s correlation analysis between *HIF-1α* and *MCT4* gene expressions in the human glioma microarray dataset GSE4290 (r = 0.2501, *p* < 0.001).

**Figure 6 cancers-12-00380-f006:**
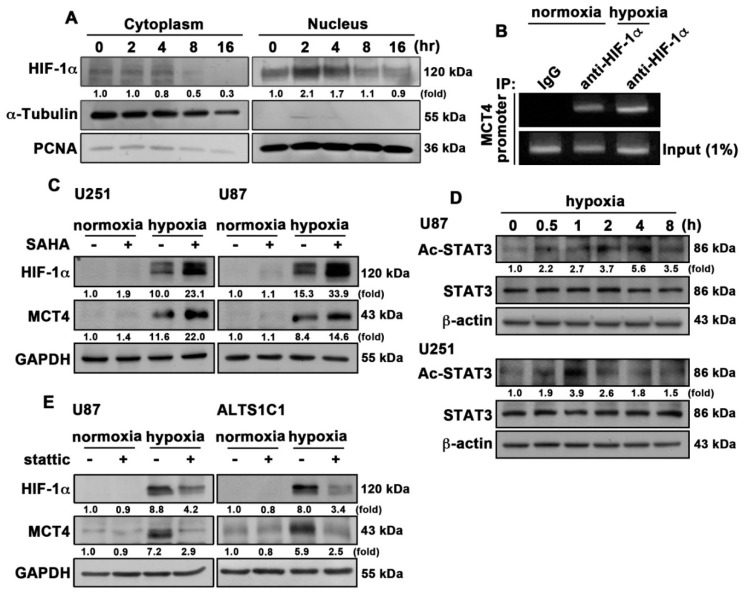
Acetylated STAT3 is involved in hypoxia-induced HIF-1α and MCT4 expressions in GBM. (**A**) The cytosolic and nuclear extracts from hypoxia-induced (1% O_2_; 2, 4, 8 or 16 h) U251 cells were subjected to western blotting. Levels of HIF-1α were determined. (**B**) U87 was exposed to hypoxic conditions (1% O_2_) for 4 h, and cells were then fixed and subjected to chromatin immunoprecipitation assay using antibodies against HIF-1α or mouse IgG. Levels of immunoprecipitated chromatin fragments of the MCT4 promoter or input were examined by PCR. (**C**) U251 (left panel) and U87 (right panel) cells were treated with SAHA (HDAC inhibitor; 2 μM) for 30 min and were exposed to hypoxic conditions (1% O_2_) for 24 h. HIF-1α and MCT4 expression levels were determined using western blotting. (**D**) U87 and U251 cells were exposed to hypoxic conditions (1% O_2_) for indicated time periods (0.5, 1, 2, 4 or 8 h), and acetylated-STAT3 expression was determined using western blotting. (**E**) U87 (left panel) and ALTS1C1 (right panel) cells were treated with stattic (STAT3 inhibitor; 10 μM) for 30 min and were exposed to hypoxic conditions (1% O_2_) for 24 h. HIF-1α and MCT4 expression levels were determined using western blotting. Quantitative data are presented as the mean ± S.E.M. (*n* = 3).

**Figure 7 cancers-12-00380-f007:**
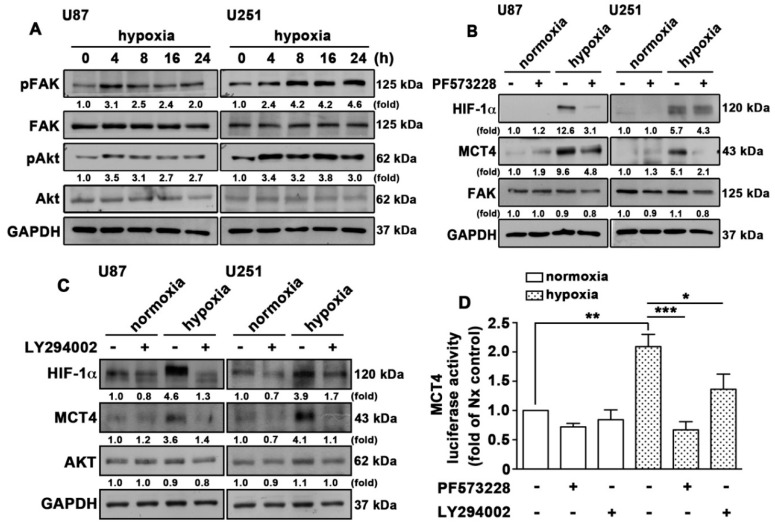
Activation of FAK and AKT is involved in the hypoxia-induced HIF-1α and MCT4 expression in GBM. (**A**) U87 and U251 cells were exposed to hypoxic conditions (1% O_2_) for indicated time periods (4, 8, 16 or 24 h), and expression of p-FAK (Tyr397) and p-AKT (Ser473) was determined using western blotting. (**B**) U87 (left panel) and U251 (right panel) cells were treated with PF573228 (2 μM) for 30 min and were exposed to hypoxic conditions (1% O_2_) for another 24 h. FAK, HIF-1α, and MCT4 expression was determined using western blotting. (**C**) U87 (left panel) and U251 (right panel) cells were treated with LY294002 (10 μM) for 30 min and were exposed to hypoxic conditions (1% O_2_) for another 24 h. AKT, HIF-1α, and MCT4 expression was determined using western blotting. (**D**) U251 cells were transfected with MCT4 luciferase reporter plasmids for 24 h and were treated with PF573228 (2 μM) or LY294002 (10 μM) for 30 min and were then exposed to hypoxic conditions (1% O_2_) for another 24 h. Firefly luciferase activity was measured, and the results were normalized to renilla luciferase activity. Two-way ANOVA with a post-hoc Bonferroni test was used to examine the significance of the mean. * *p* < 0.05 compared with the hypoxia-only group. ** *p* < 0.01 compared with the normoxia control group. *** *p* < 0.001 compared with the hypoxia-only group. Quantitative data are presented as the mean ± S.E.M. (*n* = 3).

**Figure 8 cancers-12-00380-f008:**
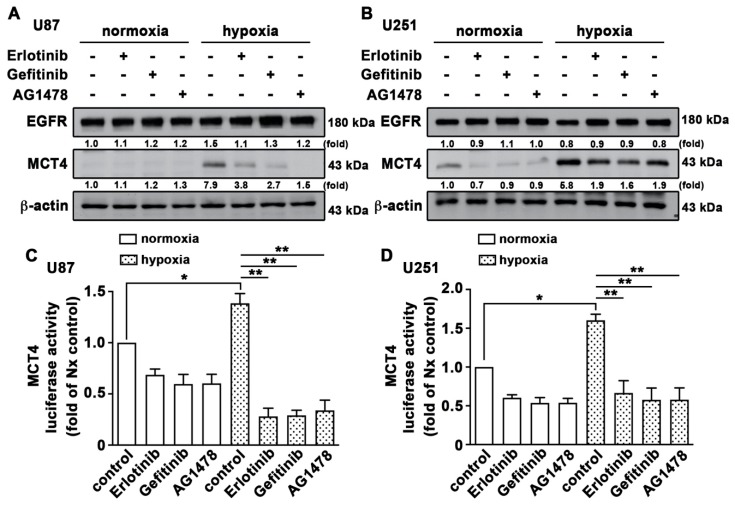
Activation of EGFR is involved in hypoxia-induced HIF-1α and MCT4 expression in GBM. U87 (**A**) and U251 (**B**) cells were treated with EGFR inhibitors (50 nM erlotinib; 50 nM gefitinib; 10 μM AG1478) for 30 min and were exposed to hypoxic conditions (1% O_2_) for another 24 h. EGFR and MCT4 expression was determined using western blotting. U87 (**C**) and U251 (**D**) cells were transfected with MCT4 luciferase reporter plasmids for 24 h and were treated with EGFR inhibitors (50 nM erlotinib; 50 nM gefitinib; 10 μM AG1478) for 30 min and were then exposed to hypoxic conditions (1% O_2_) for another 24 h. Firefly luciferase activity was measured, and the results were normalized to renilla luciferase activity. Two-way ANOVA with a post-hoc Bonferroni test was used to examine the significance of the mean. * *p* < 0.05 compared with normoxia control group. ** *p* < 0.01 compared with the hypoxia-only group. Quantitative data are presented as the mean ± S.E.M. (*n* = 3).

**Figure 9 cancers-12-00380-f009:**
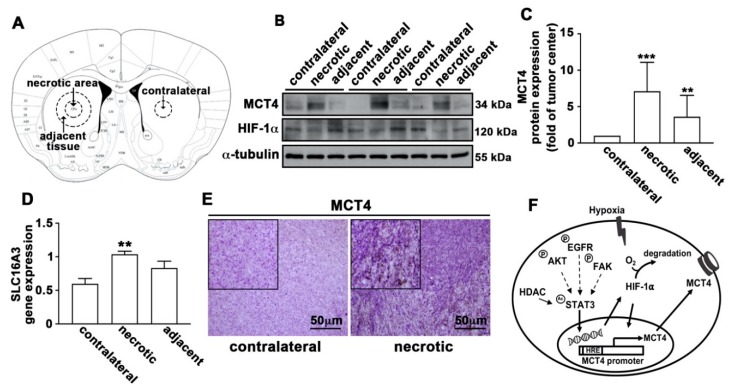
MCT4 expression is up-regulated in the GBM mouse model. ALTS1C1 (mouse GBM) cells were intracranially injected into the brain of C57BL/6 mice (*n* = 9). After 14 days, all the animals were sacrificed, and body weight was measured by using an electronic balance every other day until day 14. (**A**) The figure shows contralateral, necrotic, and adjacent tissues in the GBM mice model. MCT4 and HIF-1α expression was determined using western blotting (**B**) and real-time PCR (**D**) in the GBM of ALTS1C1-bearing mice. (**C**) Quantification of MCT4 protein levels after western blotting analysis. One-way ANOVA with a post-hoc Bonferroni test was used to examine the significance of the mean. ** *p* < 0.01 compared with the tumor contralateral group. *** *p* < 0.001 compared with contralateral tumor group. Quantitative data are presented as the mean ± S.E.M. (*n* = 3). (**E**) Immunohistochemistry with anti-MCT4 antibody in tumor tissue of GBM mice. (F) Schematic diagram showing the signaling pathways through which MCT4 modulates the microenvironment of GBM.

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
