# Peer review of "Monocarboxylate Transporter 4 Regulates Glioblastoma Motility and Monocyte Binding Ability"

_cancers, 2020, doi:10.3390/cancers12020380_

Round 1

Reviewer 1 Report

Please add a scale bar on all of the fluorescent images (Figure 2A, Figure 3A, Figure 4A). Please add a molecular weight in all of the western blot analysis images  Please show a value that quantification of band intensity in all of the western blot analysis.  line228- 231 FIg7 is not correct. It should be Fig8 please check text. In Fig8 Erlotinib, Gefitinib and AG1478 slightly decreased the expression of MCT4hypoxia condition, However, these agents significantly decreased the activity of MCT4 in Luc assay. Please explain this mechanism.

Author Response

Reply to Reviewer #1:

Major points:

Please add a scale bar on all of the fluorescent images (Figure 2A, Figure 3A, Figure 4A).

We appreciate the reviewer’s suggestions. Following the advice, we have added the scale bars in Figure 2A, Figure 3A, and Figure 4A.

Please add a molecular weight in all of the western blot analysis images. Please show a value that quantification of band intensity in all of the western blot analysis.

We are grateful to the reviewer for the comments. We have added the molecular weights in the western blot analysis images and showed the quantitative results for all of western blot analysis.

Line228-231 Figure 7 is not correct. It should be Figure 8 please check text.

We appreciate the reviewer’s comments and apologize for the typographical errors, which have been corrected on page 11, lines 231 and 232.

In Figure 8 Erlotinib, Gefitinib and AG1478 slightly decreased the expression of MCT4 hypoxia condition, However, these agents significantly decreased the activity of MCT4 in Luc assay. Please explain this mechanism.

We thank to the reviewer for expressing this concern.

We have previously observed that the differential expression of EGFR levels in GBM cells (Liu YS and Lu DY et al., Oncogene, 2017). The present study supports the previous findings that there are differential response and sensitivity of GBM cells under stimulations.

As shown in Figure 8A, U87 cells exposed to hypoxic conditions that showed approximately 7.9-fold increase in MCT4 expression. However, treatment of various of EGFR tyrosine kinase inhibitors decreased the expression of MCT4 to 3.8-fold (erlotinib), 2.7-fold (gefitinib), and 1.5-fold (AG1478) as compared with the hypoxia alone group (Figure 8A). In addition, U251 cells exposed to hypoxic conditions showed approximately 5.8-fold increase in MCT4 expression, whereas cell exposure to EGFR tyrosine kinase inhibitors decreased MCT4 expression by 1.9-fold (erlotinib), 1.6-fold (gefitinib), and 1.9-fold (AG1478) compared with that in the hypoxia alone group (Figure 8B).

On the other hand, we used the luciferase reporter assay to further detect MCT4 activation after inhibition by EGFR inhibitors. Similarly, treatment with various EGFR inhibitors significantly reduced hypoxia-enhanced MCT4 transcription activity in GBM cells (Figure 8C and 8D). Of note, U87 cells also showed higher response and sensitivity in MCT4 luciferase activity as compared to U251 cells.

Our results also suggest that there are differential limitations between these two methodologies.

Reviewer 2 Report

In the manuscript cancers-673857 that is entitled "Monocarboxylate transporter 4 regulates glioblastoma motility and monocyte binding ability," the authors aimed at dissecting the biological functions of Monocarboxylate Transporter 4 (MCT4) in glioma cells under hypoxic coditions. The authors showed that the association between HIF-1a and MCT4 in glioma (Figs 1, 5, 9) and their functional impacts (Figs 2-4). Moreover, the authors showed the association of STAT3 (Fig 6), AKT/FAK (Fig 7), and EGFR (Fig 8). Collectively, the authors claim that MCT4-regulated GBM motility and monocyte adhesion are mediated by activation of AKT, FAK and EGFR signaling pathways. This reviewer has several comments regarding this study.

1. The authors should illustrate their working hypothesis based on their findings as Fig 9. Once the authors illustrate the working hypothesis, they may reorder the figures for better understandability.

2. Since the STAT3 directly activates gene expression, it's reasonable to observe the impact of the EGFR-to-STAT3 pathway on the expression changes of HIF-1a and MCT4. However, there seems to be a missing link regarding the AKT/FAK pathway.The authors should explain more how this pathway relates to the other pathways the authors presented in this study.

3. Regarding statistics, there are a number of inconsistencies in the Methods section and the figure legends. For instance, the authors stated that they used Student's t-test for multiple comparisons in Figs 1E, 2D, 5B,5D, and some more.

4. Some important abbreviations are missing such as AKT, FAK, STAT3, and EGFR.

5. There are a good number of grammatical errors that should be corrected.

Author Response

Reply to Reviewer #2:

Reviewer #2 (Comments and Suggestions for Authors):

In the manuscript cancers-673857 that is entitled "Monocarboxylate transporter 4 regulates glioblastoma motility and monocyte binding ability," the authors aimed at dissecting the biological functions of Monocarboxylate Transporter 4 (MCT4) in glioma cells under hypoxic coditions. The authors showed that the association between HIF-1a and MCT4 in glioma (Figs 1, 5, 9) and their functional impacts (Figs 2-4). Moreover, the authors showed the association of STAT3 (Fig 6), AKT/FAK (Fig 7), and EGFR (Fig 8). Collectively, the authors claim that MCT4-regulated GBM motility and monocyte adhesion are mediated by activation of AKT, FAK and EGFR signaling pathways. This reviewer has several comments regarding this study.

Major points:

The authors should illustrate their working hypothesis based on their findings as Fig 9. Once the authors illustrate the working hypothesis, they may reorder the figures for better understandability.

We really appreciate the reviewer’s suggestion. Following the advice, we have added an illustration of our working hypothesis in Figure 9A, and reordered the figures.

Since the STAT3 directly activates gene expression, it's reasonable to observe the impact of the EGFR-to-STAT3 pathway on the expression changes of HIF-1a and MCT4. However, there seems to be a missing link regarding the AKT/FAK pathway. The authors should explain more how this pathway relates to the other pathways the authors presented in this study.

We appreciate the reviewer’s suggestion. We have added a discussion of AKT and FAK signaling pathway in the Discussion section. (Pages 13 and 14)

High level of phosphorylated AKT has been reported to correlate with poor prognosis for patients with GBM. HIF-1α is downstream of EGFR and AKT signaling and activates VEGF expression in tumor cells. In addition, our previous study also supports the finding that hypoxia-induced iNOS expression in microglia is HIF-1α-dependent and involves the activation of PI3-kinase/AKT/mTOR signaling pathway. Another study revealed that AKT inhibitors decrease HIF-1α post-transcriptional protein levels, but not the transcriptional mRNA levels in the brain tissue. It has been reported that FAK is typically considered upstream of AKT, and extracellular pressure stimulates cancer cell adhesion via AKT-dependent FAK activation. A previous study has demonstrated that overactivation of STAT3 is conducive to tumor invasion through the up-regulation of FAK in gastric cancer cells. Our present study also found that FAK/STAT3 signaling is involved in GBM migration and IL-8 production. It has been reported that coactivation of EGF and EGFR drives tumor metastasis via PI3K/AKT-dependent pathways. Targeting FAK/STAT3 sensitizes gliomas to anti-EGFR agent gefitinib and alkylating agent temozolomide in human gliomas. Our previous study has shown that STAT3 signaling is involved in the EGFR associated adhesion molecule expression and monocyte adhesion in GBM. In this study, we suggest that MCT4-regulated GBM motility and monocyte adhesion are mediated by activation of AKT, FAK, and EGFR signaling pathways. However, further research is required for exploring the specific mechanism by which AKT, FAK, and EGFR/STAT3 affect the expression of MCT4, and to determine how pH-regulating proteins affect GBM in a suppressive immune microenvironment.

Regarding statistics, there are a number of inconsistencies in the Methods section and the figure legends. For instance, the authors stated that they used Student's t-test for multiple comparisons in Figs 1E, 2D, 5B,5D, and some more.

We appreciate the reviewer’s suggestion. Following the comments, we have rechecked and revised the Method section and all the figure legends.

Some important abbreviations are missing such as AKT, FAK, STAT3, and EGFR.

All the abbreviations used in the manuscript have been checked thoroughly. (Pages 2 and 10)

There are a good number of grammatical errors that should be corrected.

We really appreciate the reviewer’s comments. The grammar of the paper has been corrected thoroughly and checked by an English academic.

Round 2

Reviewer 2 Report

Although the authors tried to address this reviewer's comments, they appear to lack fundamental knowledge of statistics.

1. For multiple group comparison, one can never use t-test. Instead, ANOVA with post-hoc test should be conducted. Therefore, most of the statistical interpretation in this study were incorrectly conducted.

2. The Fig 9A just illustrates the in vivo experiments. This reviewer suggested the authors to illustrate a schema of all the signal pathways involved in this study.

Author Response

Comments and Suggestions for Authors

Although the authors tried to address this reviewer's comments, they appear to lack fundamental knowledge of statistics.

1. For multiple group comparison, one can never use t-test. Instead, ANOVA with post-hoc test should be conducted. Therefore, most of the statistical interpretation in this study were incorrectly conducted.

Ans:

1). We thank the reviewer for the constructive suggestions., we have re-analyzed the quantitative results and statistical analysis by Bonferroni correction. The data given are statistical analysis between two samples that were performed using a Student’s t-test. One-way ANOVA and Two-way ANOVA (GBM treatment with drugs under hypoxic conditions) assessed statistical analysis between three or more independent group was used to determine. P-value < 0.05 was considered to be of statistical significance.

2. The Fig 9A just illustrates the in vivo experiments. This reviewer suggested the authors to illustrate a schema of all the signal pathways involved in this study.

Ans:

2). We thank the reviewer for the suggestion. We have illustrated a schematic diagram of the signaling pathways in figure 9F.

Schematic diagrams of the signaling pathways are involved in MCT4 modulates microenvironment of GBM. MCT4-regulated GBM cell motility and monocyte adhesion are mediated by activation of AKT, FAK, and EGFR signaling pathways. Moreover, hypoxia mediated the acetylated STAT3 expression and regulated the transcriptional activity of HIF-1α in GBM. (Page 12)

Round 3

Reviewer 2 Report

In the revised manuscript cancers-673857-v3, the authors correctly conducted statistical analyses and also added informative schematic diagram as Fig 9F to explain their working hypothesis. Since the newly added descriptions have a good number of grammatical errors, the manuscript should undergo grammatical proofreading once again. Other than it, the authors addressed this reviewer's comments almost appropriately.

Author Response

We thank for the reviewer's suggestion.

English has been checked by an English academic and the certificate has been attached below.

CERTIFICATE OF
ENGLISH EDITING
This document certifies that the paper listed below has been edited to ensure that the
language is clear and free of errors. The edit was performed by professional editors at Editage,
a division of Cactus Communications.The intent of the author's message was not altered in any
way during the editing process. The quality of the edit has been guaranteed, with the
assumption that our suggested changes have been accepted and have not been further altered
without the knowledge of our editors.
TITLE OF THE PAPER
Monocarboxylate transporter 4 regulates glioblastoma motility and monocyte binding
ability
AUTHORS
Dah-Yuu Lu
JOB CODE
WOPFK_19
Editage, a brand of Cactus Communications, of f ers prof essional English language
editing and publication support services to authors engaged in over 500 areas of
research. Through its community of experienced editors, which includes doctors,
engineers, published scientists, and researchers with peer review experience,
Editage has successf ully helped authors get published in internationally reputed
journals. Authors who work with Editage are guaranteed excellent language quality
and timely delivery.
Contact Editage

Signature
Vikas Narang,
Chief Operating Of f icer,
Editage
Date of Issue
January 09, 2020
Worldwide
[email protected]
+1 877- 334- 8243
www.editage.com
Japan
[email protected]
+81 03- 6868- 3348
www.editage.jp
Korea
submitkorea@
editage.com
1544- 9241
www.editage.co.kr
China
[email protected]
400- 005- 6055
www.editage.cn
Braz il
[email protected]
0800- 892- 20- 97
www.editage.com.br
Taiwan
[email protected]
02 2657 0306
www.editage.com.tw
